genetics

chickens, mitochondrial DNA, South America, phylogeography

# European and Asian contribution to the genetic diversity of mainland South American chickens

Michael B. Herrera[1,2,†], Spiridoula Kraitsek[3,†], Jose A. Alcalde[4], Daniel Quiroz[5], Herman Revelo[6], Luz A. Alvarez[6], Millor F. Rosario[7], Vicki Thomson[1], Han Jianlin[8,9], Jeremy J. Austin[1] and Jaime Gongora[3]

[1]Australian Centre for Ancient DNA, School of Biological Sciences, University of Adelaide, Adelaide, Australia
[2]Archaeological Studies Program, University of the Philippines Diliman, Quezon City, Philippines
[3]Sydney School of Veterinary Science, Faculty of Science, University of Sydney, Sydney, Australia
[4]Facultad de Agronomia e Ingenieria Forestal, Pontificia Universidad Catolica de Chile, Santiago, Chile
[5]Dirección de Bibliotecas, Archivos y Museos-Proyecto Fondecyt, Santiago, Chile
[6]Departamento de Ciencia Animal, Universidad Nacional de Colombia, sede Palmira, Colombia
[7]Nature Science Center, Federal University of São Carlos, São Carlos, Brazil
[8]CAAS-ILRI Joint Laboratory on Livestock and Forage Genetic Resources, Institute of Animal Science, Chinese Academy of Agricultural Sciences (CAAS), Beijing, People's Republic of China
[9]Livestock Genetics Program, International Livestock Research Institute (ILRI), Nairobi, Kenya

 MBH, 0000-0001-6548-0816; VT, 0000-0001-8368-9664; JJA, 0000-0003-4244-2942; JG, 0000-0003-2215-656X

**Author for correspondence:**
Jaime Gongora
e-mail: jaime.gongora@sydney.edu.au

[†]Contributed equally to this work.

Chickens (*Gallus gallus domesticus*) from the Americas have long been recognized as descendants of European chickens, transported by early Europeans since the fifteenth century. However, in recent years, a possible pre-Columbian introduction of chickens to South America by Polynesian seafarers has also been suggested. Here, we characterize the mitochondrial control region genetic diversity of modern chicken populations from South America and compare this to a worldwide dataset in order to investigate the potential maternal genetic origin of modern-day chicken populations in South America. The genetic analysis of newly generated chicken mitochondrial control region sequences from South America showed that the majority of chickens from the continent belong to mitochondrial haplogroup E. The rest belongs to haplogroups A, B and C, albeit at very low levels.

Haplogroup D, a ubiquitous mitochondrial lineage in Island Southeast Asia and on Pacific Islands is not observed in continental South America. Modern-day mainland South American chickens are, therefore, closely allied with European and Asian chickens. Furthermore, we find high levels of genetic contributions from South Asian chickens to those in Europe and South America. Our findings demonstrate that modern-day genetic diversity of mainland South American chickens appear to have clear European and Asian contributions, and less so from Island Southeast Asia and the Pacific Islands. Furthermore, there is also some indication that South Asia has more genetic contribution to European chickens than any other Asian chicken populations.

## 1. Introduction

The domestic chicken (*Gallus gallus domesticus*) is the world's most ubiquitous and important bird species. It represents one of the main sources of animal protein around the world, thus it plays a major role in global food security. Furthermore, the socio-cultural role of the domestic chicken cannot be overemphasized [1]. Despite this, our understanding of the chicken domestication process and global translocation history remains incomplete. Chicken domestication potentially happened in several episodes involving one or more wild progenitor species across a wide geographical region. Domestication probably involved the selection of desired traits and behaviour from a wild red junglefowl (RJF) progenitor (*Gallus gallus*). However, there are indications that this process also included hybridization of the red and grey junglefowls (*Gallus sonneratti*) [2]. This complexity reflects the uncoordinated nature of many domestication events performed by early human cultures. The subsequent translocation of the domestic chicken out of their domestication centres is nuanced by the protracted and complex movements (diaspora and trade) of humans across the globe, including those during the modern era.

The ultimate origin of today's European domestic chickens is somewhere within the natural biogeographic range of junglefowls (i.e. *G. gallus* and *G. sonneratti*). This includes domestication centres within South and Southeast Asia [3], and potentially China [4]. A recent bioarchaeological study suggests China (as a region) represents the earliest site for chicken domestication [5], although this is controversial [6]. It is suggested that chickens reached Europe via trading networks from Asia either north through China and Russia [4] or south through Mesopotamia to Greece then westwards towards the rest of Europe [7]. The economic exploitation of chickens is apparent between the fourth and second centuries AD in the Southern Levant [8]. Archaeological chicken remains from Central Asia seem to indicate that chickens arrived in Europe around 3000 BC. [4]. Domestic chickens then became well established in Europe during the Iron Age [9]. These early European chickens belong to mitochondrial haplogroup E [10], a genetic lineage that is ubiquitous today in the Indian subcontinent [3,11,12]. In Africa, the process and timing of domestic chicken introductions is less clear [13]. However, it is likely that East African chickens also originated directly from the Indian subcontinent [14,15]. From Europe, chickens are believed to have been brought to the New World during the early contact periods. However, a pre-Columbian Polynesian introduction of chickens to the New World has been recently hypothesized [16–19]. By contrast, chickens from the Pacific are thought to have a southeast Asian origin [20,21].

The potential for human interactions between Polynesia and South America during pre-Columbian times have long been a source of interest and controversy. These interactions potentially facilitated the translocations of species from the Pacific into South America and vice versa. Some species used to examine the interactions between Polynesia and South America include the bottle gourd (*Lagenaria siceraria*) [22,23], sweet potato (*Ipomoea batatas*) [24–28] and domestic chicken [18,19,29,30]. While some studies of these species have been used to infer a pre-Columbian contact between Polynesia and South America, other studies dispute this [31–33]. A recent genome-wide study of sweet potato questions the existence of this contact [34]. Notwithstanding this debate, DNA analyses on translocated species remain helpful in studying movements where gene flow between human populations is minimal, absent, hard to study [16,35], or when archaeological evidence is unavailable.

Chickens from the Americas have long been considered as descendants of European chickens, brought by the early Europeans since the fifteenth century [7]. However, historical accounts describe a high degree of integration of chickens into Incan culture at the onset of European contact [36,37], suggesting an earlier introduction of chickens into South America. A pre-Columbian introduction from Polynesia has been suggested based on chicken mitochondrial DNA (mtDNA) [18]. However, subsequent work examining the relationships of continental South American and Pacific chickens challenges this conclusion [20,33].

Radiocarbon dates have also been used to suggest a pre-Columbian introduction of chickens to South America [18], though there is also debate over the reliability of these radiocarbon dates [33]. The presence of mtDNA haplogroup D in an early post-European Peruvian specimen has also been used to suggest chickens from this country may have originated from Polynesia during pre-Colombian times [19]. Thus, the Americas may have experienced at least two translocations of chickens, initially by the Polynesians and subsequently by Europeans [19]. Haplotype E1 is ubiquitous worldwide and considered phylogeographically uninformative [33] and its presence in ancient Polynesian samples is suggested to be a result of laboratory contamination [20]. These issues have been discussed extensively in the literature [16,38,39].

A recent study comparing contemporary chickens from South America and the Iberian Peninsula (Spain and Portugal) suggests that the observed genetic differentiation between the two regions is due to another (unsourced Asian) maternal source for South American chickens [40]. That study indicates that despite the global movement of chicken during modern times, the genetic patterns from the initial translocation can still be inferred.

In this study, we extend both the South American and comparative sampling of the previous study to characterize the contemporary mtDNA control region (CR) DNA data from South America and compare to other chicken populations from across the globe (from Europe to Island and Mainland Southeast Asia, East Asia, the Pacific Islands, South Asia and Southwest Asia). We assess the ancestry of modern South American chickens as a potential way to infer the colonization history of the continent by Europeans and later trade networks with Asia.

# 2. Methods

## 2.1. Chicken samples, polymerase chain reaction and sequencing

Blood samples were collected from a total of 229 native chickens from four South American countries (excluding Easter Island, which although it is a special territory of Chile, is considered to be culturally aligned with the Pacific region): 30 from Brazil, 60 from Chile, 129 from Colombia and 10 from Peru. Blood samples were collected from the brachial vein of the wing and transferred to FTA cards (Qiagen, Hilden Germany). DNA was extracted using QIAamp DNA Investigator Kit (Qiagen, Hilden, Germany). The mtDNA CR was chosen as the target as it is highly polymorphic and phylogeographically informative [3,14,20,33,41]. The target region of mitochondrial hypervariable region 1 was amplified using the following primer set: 5′-AGGACTACGGCTTGAAAAGC-3′ and 5′-ATGTGCCTGACCGAGGAACCAG-3′. DNA was amplified using polymerase chain reaction (PCR) in 30 µl reaction volumes containing 50 mM KCl, 10 mM Tris–HCl (pH 8.3), 0.1% Triton X-100, 1.5 mM MgCl$_2$, 0.2 mM dNTPs, 0.1 µM concentrations of each primer, 1.25 units of *Taq* DNA polymerase (Promega) and 100–200 ng of template DNA. PCR cycling condition included an initial denaturation of 94°C for 2 min, followed by 35 cycles of 25 s at 94°C, 35 s at 58°C, and 1 min 10 s at 72°C, and a final extension for 10 min at 72°C. Sanger sequencing was conducted at the Australian Genome Research Facility Ltd (AGRF) in Brisbane. The raw forward and reverse chromatograms were assembled, edited and inspected by eye to give a consensus sequence of a 530 bp fragment for each sample excluding primer sequences.

## 2.2. Sequence data, phylogenetic and population genetic analyses

In addition to the 229 control region sequences generated in this study, we included 2618 worldwide mtDNA control region sequences from GenBank to examine the relationship of mainland South American chickens with those from South Asia (India, Bangladesh), Mainland Southeast Asia (MSEA: Laos, Myanmar, Thailand, Vietnam), Island Southeast Asia (ISEA: Indonesia, Philippines), Pacific Islands (Fiji, Solomon Islands, Vanuatu, Easter Island), Central Asia (Azerbaijan, Turkmenistan), East Asia (China, Korea) and Europe (electronic supplementary material, table SI 1). A total of 2827 sequences were aligned using the MUSCLE [42] algorithm in Geneious v. 11.0.4 [43] and trimmed to produce a final 412 bp fragment corresponding to mtDNA CR positions 93–504 of the reference sequence NC_007235 [44]. Truncation of the new sequences to the 412 bp fragment was made to directly compare with the South American chicken samples from Luzuriaga-Neira *et al.* [40]. The number and assignment of haplotypes of the 412 bp CR dataset was determined using DnaSP v. 6 [45]. The haplogroup of the newly generated sequences were established by comparison with

sequences of known haplogroup designation [3,12]. This was executed using a combination of neighbour-joining (NJ) and median-joining (MJ) [46] analyses. jModelTest [47] was used to determine the best-fit model for the CR dataset (TIM1 + G), this was performed through the CIPRES Science Gateway [48], then an NJ tree was estimated using PAUP v. 4 [49]. The phylogenetic structure of the South American chicken sequences used in this study was also characterized by using the network analysis (MJ) implemented in PopART v. 1.7.1 [50]. The program Haplotype Viewer (http://www.cibiv.at/~greg/haploviewer) was also used to visualize the haplotype genealogies for the whole dataset.

The population genetic structure among the sampling locations was estimated using Slatkin's linearized $F_{ST}$ as implemented in Arlequin v. 3.5.2.2 (10 000 permutations) [51]. To visualize the relationships of the sampling populations, the $F_{ST}$ scores were ordered into principal coordinate analysis (PCoA) plots using GenAlEx v. 6.503 [52]. This analysis was initially performed for all populations included in the study. By removing the outliers in PC1 of this PCoA plot (responsible for approx. 30% of variation explained by PC1), we generated a second PCoA plot to investigate which geographical regions the South American chickens are most allied with. Population genetic structure was further investigated using analysis of molecular variance (AMOVA) as implemented by Arlequin v. 3.5.2.2 [51]. The groupings used in the AMOVA compared chicken populations from seven regions including South America, ISEA-Pacific, MSEA, South Asia, Europe and Central Asia. The different population hypotheses were tested initially using the overall dataset assuming no groups and hierarchically comparing populations from different geographical regions indicated previously. Significance testing was done using 10 000 coalescent simulations in Arlequin v. 3.5.2.2 [51]. Intra-population genetic variability statistics (i.e. segregating sites, number of haplotypes, haplotype and nucleotide diversities) were also calculated using DnaSP v. 6 [45].

# 3. Results and discussion

A previous mtDNA study revealed nine divergent haplogroups (A–I) of chickens from across the world [3]. A more fine-grained mtDNA genome phylogeny study revealed an additional four haplogroups (W–Z) [12]. Haplogroup A and B are predominantly found among southern and eastern Chinese and Japanese chickens as well as wild RJF. Haplogroup C is found mainly in Japanese and southeast Chinese chickens. Haplogroup D is found in Japanese, southeast Chinese, Mainland Southeast Asian and Pacific chickens. Haplogroup E is widespread among Indian, Middle Eastern and European chickens.

Five of these haplogroups (A–E) are relevant to the present study as they are found in South American chickens (figure 1). Haplogroup D has not been found in modern South American chickens in this study. Haplogroup E is the most predominant mitochondrial CR region lineage observed in South America comprising 83.96% of all chickens on the continent (table 1). Haplogroups A (7.35%), B (4.68%) and C (4.00%) are also observed in modern South American chickens, albeit at very low frequencies. All four of these haplogroups are related to those found in Asian and European chicken populations.

Haplogroup E is also the most dominant haplogroup observed across each of the South American countries we studied (i.e. Bolivia, Brazil, Chile, Ecuador and Peru). Given the high frequency of this haplogroup in Europe along with the historical records and observations that Spanish and Portuguese brought chickens to the Americas [36], Europe may be the more likely source of modern chickens in South America. The most ubiquitous haplogroup E lineage in South America is haplotype 107 (haplotype E1; figure 2). It is observed in all the South American populations in this study. This haplotype potentially represents the founding haplotype brought by the Europeans. Furthermore, haplotype 107 is the lineage observed in archaeological chickens in Europe [10].

The presence of haplogroups A, B and C in South American chickens could represent subsequent introductions from Asian into the South American populations. Haplogroup D profiles are not observed in modern South American chickens from the mainland. Rather, haplogroup D is the dominant haplogroup in the Pacific Island [20] and Island Southeast Asian [14] chickens. Thus, this potentially indicates that the translocation history of haplogroup D chickens from Island Southeast Asia into the Pacific islands did not include the successful introduction of Polynesian chickens into continental South America.

The PCoA analyses (figures 3 and 4) shows that the genetic relationship of South American chickens is largely allied with European populations in comparison to other parts of the world. In particular, this affinity is more pronounced with chickens from the Iberian Peninsula (i.e. Portugal and Spain). The

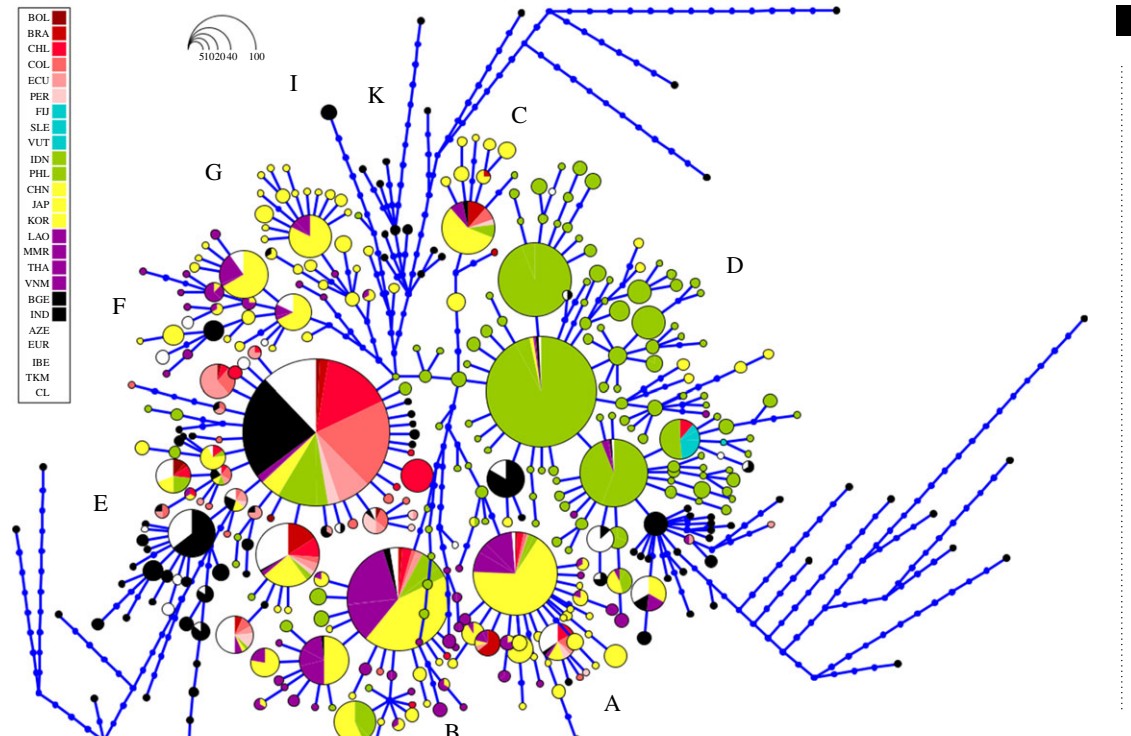

**Figure 1.** Network of the mitochondrial control region haplotypes found in South America, the Pacific, Island Southeast Asia, Mainland Southeast Asia, East Asia, South Asia, Europe and Central Asia. The size of the circles is proportional to the frequency of each haplotype. Mutational steps between haplotypes are represented by small blue dots, linked by blue lines. The abbreviations for chicken populations are as follows: BOL, Bolivia; BRA, Brazil; CHL, Chile; COL, Colombia; ECU, Ecuador; PER, Peru; FIJ, Fiji; SLE, Solomon; VUT, Vanuatu; IDN, Indonesia; PHL, Philippines; CHN, China; JAP, Japan; KOR, Korea; LAO, Laos; MMR, Myanmar; THA, Thailand; VNM, Vietnam; BGE, Bangladesh; IND, India; AZE, Azerbaijan; EUR, Europe; IBE, Iberia; TKM, Turkmenistan; and CL, Commercial Lines.

geographical group of chickens with the next closest affinity to those in South America are South Asian chickens, then East Asian and Southeast Asia. It can also be observed that South American chickens are only remotely related to chickens from the Pacific, despite their geographical proximity (figure 3). In general, the modern diversity of South American chickens does not appear to support a pre-Colombian Polynesian origin. If a pre-Columbian introduction of chickens to South America did happen, its genetic signature did not persist through to modern times. Furthermore, the close relationship of European chickens to those from South Asia seems to potentially suggest a history based on initial domestication in that region and subsequent translocation(s) to Europe. The archaeological documentation of chicken remains from the Indus Valley suggests that South Asia was indeed one of the domestication centres for chickens [7]. The cultural contact, trade and migration of this early civilization probably allowed for the human-mediated transport of chickens from South Asia to Western Asia, through the Arabian peninsula and then on to the Mediterranean region [7]. One of these trajectories could also have involved the movement and introduction of chickens to the east coast of Africa [14].

The PCoA plot (figure 4) also indicates that Iberian chickens are more closely related to chicken populations from the western side of the South American continent (i.e. Columbia, Chile, Ecuador) than the eastern side (i.e. Brazil). This could potentially reflect the trading network between Spain, Portugal, South America and the Orient. The initial European voyages into South America involved a long-distance trans-Atlantic crossing from the Iberian Peninsula arriving at the West Indies or at several mainland port cities such as Veracruz in Mexico and Panama-Porto Bello. Then, trade and exchange via the land caravan or China Road (*Camino de China*) occurred between Vera Cruz on the Atlantic side and Acapulco on the Pacific side (via Puebla) [53]. Mexico City was also part of these early over-land trade routes [54]. From Acapulco, merchant ships sailed west across the Pacific to Manila via the Manila galleon trade seeking oriental products [55], but ships could also have proceeded to other ports on the western coast of South America [56]. The maritime network from Acapulco to other ports on the west coast of South America is highly likely, based on the availability

**6**

**Table 1.** Mitochondrial control region haplogroup composition of chickens from South America, Island Southeast Asia (ISEA), Pacific Islands, East Asia, Mainland Southeast Asia (MSEA), South Asia, Europe and Central Asia.

| region | populations | A | B | C | D | E | F | G | H | I | K | total |
|---|---|---|---|---|---|---|---|---|---|---|---|---|
| South America | Bolivia | 1 | 0 | 0 | 0 | 10 | 0 | 0 | 0 | 0 | 0 | 11 |
| | Brazil | 9 | 3 | 9 | 0 | 26 | 0 | 0 | 0 | 0 | 0 | 47 |
| | Chile | 9 | 9 | 0 | 0 | 119 | 0 | 0 | 0 | 0 | 0 | 137 |
| | Colombia | 8 | 4 | 5 | 0 | 124 | 0 | 0 | 0 | 0 | 0 | 141 |
| | Ecuador | 3 | 5 | 2 | 0 | 72 | 0 | 0 | 0 | 0 | 0 | 82 |
| | Peru | 3 | 0 | 2 | 0 | 26 | 0 | 0 | 0 | 0 | 0 | 31 |
| ISEA Pacific Islands | Indonesia | 9 | 42 | 0 | 541 | 32 | 0 | 0 | 1 | 0 | 0 | 625 |
| | Philippines | 3 | 19 | 6 | 179 | 52 | 0 | 0 | 0 | 0 | 0 | 259 |
| | Fiji | 0 | 0 | 0 | 2 | 0 | 0 | 0 | 0 | 0 | 0 | 2 |
| | Solomon Is. | 0 | 0 | 0 | 3 | 0 | 0 | 0 | 0 | 0 | 0 | 3 |
| | Vanuatu | 0 | 0 | 0 | 9 | 0 | 0 | 0 | 0 | 0 | 0 | 9 |
| | Easter Island | 0 | 0 | 0 | 4 | 0 | 0 | 0 | 0 | 0 | 0 | 4 |
| East Asia | China | 179 | 173 | 60 | 20 | 46 | 75 | 77 | 0 | 0 | 0 | 630 |
| | Korea | 0 | 5 | 9 | 0 | 17 | 0 | 0 | 0 | 0 | 0 | 31 |
| | Japan | 1 | 0 | 0 | 0 | 0 | 0 | 0 | 0 | 0 | 0 | 1 |
| MSEA | Laos | 17 | 39 | 0 | 6 | 1 | 4 | 2 | 0 | 0 | 0 | 69 |
| | Myanmar | 9 | 14 | 4 | 0 | 1 | 8 | 0 | 0 | 0 | 0 | 36 |
| | Thailand | 0 | 2 | 1 | 2 | 0 | 3 | 0 | 0 | 4 | 0 | 12 |
| | Vietnam | 30 | 80 | 5 | 3 | 10 | 19 | 9 | 0 | 5 | 0 | 161 |

(Continued.)

**Table 1.** (*Continued.*)

| region | populations | A | B | C | D | E | F | G | H | I | K | total |
|---|---|---|---|---|---|---|---|---|---|---|---|---|
| South Asia | India | 2 | 6 | 11 | 71 | 219 | 9 | 0 | 0 | 4 | 18 | 340 |
| | Bangladesh | 9 | 0 | 8 | 18 | 42 | 19 | 0 | 0 | 0 | 0 | 96 |
| Europe | Europe | 2 | 1 | 1 | 8 | 20 | 0 | 0 | 0 | 0 | 0 | 32 |
| | Iberia | 1 | 4 | 0 | 0 | 68 | 0 | 0 | 0 | 0 | 0 | 73 |
| Central Asia | Azerbaijan | 0 | 0 | 0 | 0 | 5 | 0 | 0 | 0 | 0 | 0 | 5 |
| | Turkmenistan | 0 | 1 | 0 | 0 | 4 | 0 | 0 | 0 | 0 | 0 | 5 |
| — | Commercial | 0 | 0 | 0 | 0 | 5 | 0 | 0 | 0 | 0 | 0 | 5 |

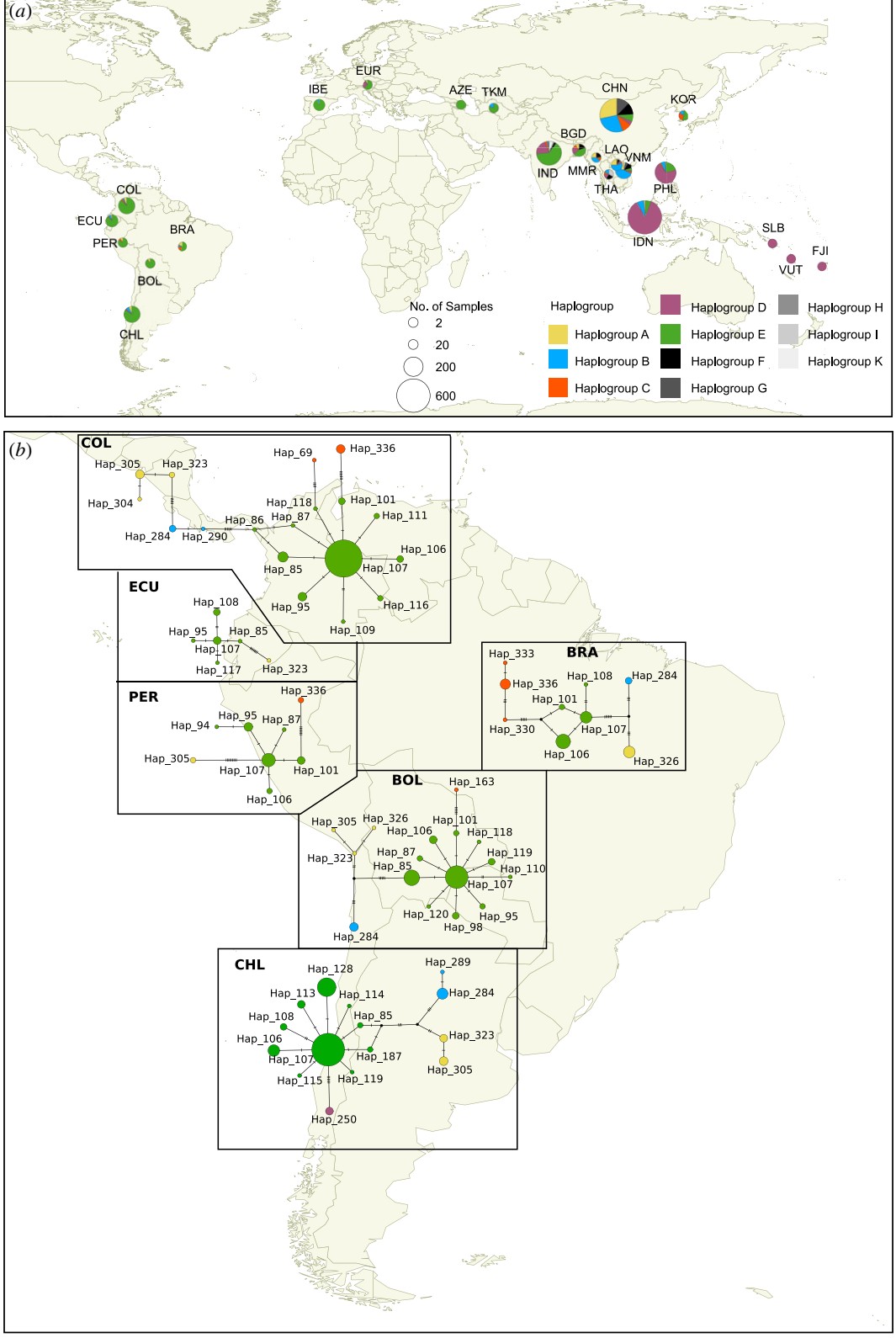

**Figure 2.** Worldwide map showing the distribution of chicken mitochondrial haplogroups. (*a*) Frequency of chicken mitochondrial haplogroups from South America: Bolivia (BOL), Brazil (BRA), Chile (CHL), Colombia (COL), Ecuador (ECU) and Peru (PER); Island Southeast Asia-Pacific: Indonesia (IDN), Philippines (PHL), Fiji (FJI), Solomon Islands (SLB), Vanuatu (VUT); East Asia: China (CHN) and Korea (KOR); Mainland Southeast Asia: Laos (LAO), Myanmar (MMR), Thailand (THA), Vietnam (VNM); South Asia: India (IND) and Bangladesh (BGD); Southwest Asia: Azerbaijan (AZE) and Turkmenistan (TKM); Europe (EUR) and Iberia (IBE). The colours in (*a*) correspond to the colours depicted in (*b*). (*b*) Median-joining network of the mitochondrial control region haplotypes found in South American populations. Haplotype 250 is from Easter Island.

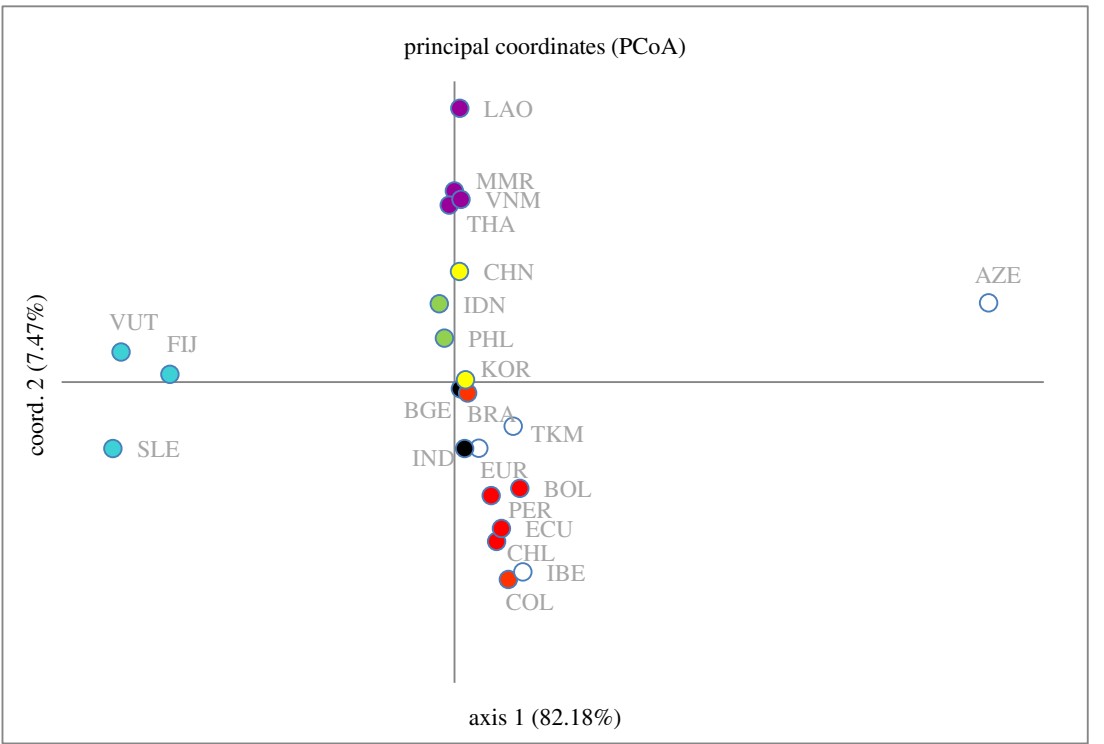

**Figure 3.** PCoA plots of population pairwise $F_{st}$ values for 2847 chicken samples worldwide using all haplogroups. Geographically based populations are assigned the following colours (red: South America; blue: Pacific; violet: Mainland Southeast Asia; yellow: East Asia; green: Island Southeast Asia; black: South Asia; and white: Europe and Southwest Asia).

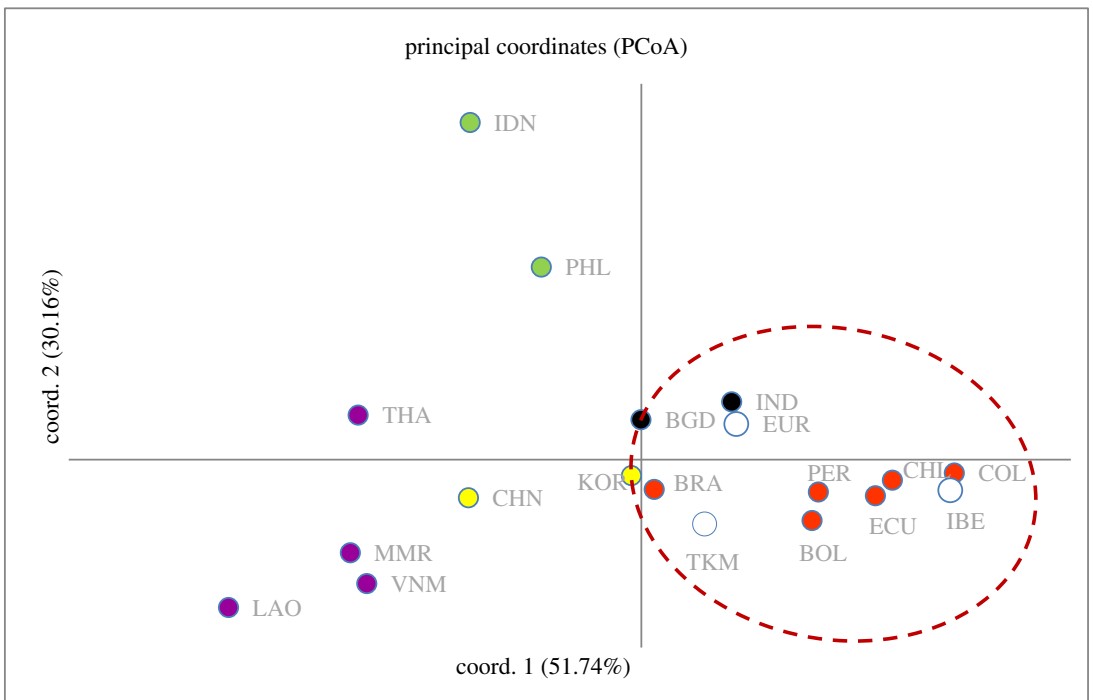

**Figure 4.** PCoA plots of population pairwise $F_{st}$ values of chickens from selected regions of the globe using all haplogroups. Populations are assigned the following colours (red: South America; violet: Mainland Southeast Asia; yellow: East Asia; green: Island Southeast Asia; black: South Asia; and white: Europe and Southwest Asia).

of precious metals (like silver from Peru), food, textiles and several other important trade goods [56]. The genetic relationship illustrated by chicken populations from Iberia and western South America could potentially reflect this scenario.

**Table 2.** Population genetic structure estimated from the analysis of molecular variance (AMOVA) based on the chicken mtDNA D-loop sequences from (1) South America, (2) ISEA-Pacific, (3) East Asia, (4) MSEA, (5) South Asia, (6) Europe and (7) Southwest Asia.

| group | no. population | no. groups | variance components (%) | | |
|---|---|---|---|---|---|
| | | | among groups | among populations within group | within populations |
| A. no grouping | 24 | 1 | — | 27.72 | 72.28 |
| B. group 1<br>1 versus 2 versus 3 versus 4 versus 5 versus 6 versus 7) | 24 | 7 | 24.79 | 4.57 | 70.63 |
| C. group 2<br>1,6,7 versus 2,3,4,5) | 24 | 2 | 12.17 | 21.15 | 66.67 |
| D. group 3<br>1,3,4,5,6,7 versus 2) | 24 | 2 | 16.02 | 17.54 | 66.44 |
| E. group 4<br>1,3,4,5 versus 2 versus 6,7) | 24 | 3 | 15.49 | 17.21 | 67.30 |
| F. group 5<br>1,2 versus 3,4,5,6,7) | 24 | 2 | 7.18 | 22.74 | 70.08 |
| G. group 6<br>1,6,7 versus 2 versus 3,4,5) | 24 | 3 | 18.61 | 12.90 | 68.48 |
| H. group 7<br>1,2,6,7 versus 3,4,5) | 24 | 2 | 7.89 | 22.24 | 69.87 |
| I. group 8<br>1,6,7,5 versus 2 versus 3,4) | 24 | 3 | 27.26 | 5.02 | 67.73 |

A pre-Columbian contact between South America and Polynesia has been suggested by other lines of evidence [22,24,25]. Some researchers hypothesize that chickens were initially introduced to South America by the Polynesians before the initial arrival of Europeans [18]. This scenario appears to be tenuous when using modern DNA chicken datasets, as South American chicken populations are clearly more related to chickens from Europe. However, this does not discount the potential for a pre-Columbian introduction of chickens by the Polynesians to South America. All we can say is that there is no evidence for Polynesian chicken genetic signatures being retained by contemporary South American chickens. Caution has to be taken in reconstructing certain aspects of past human behaviour when using only modern DNA, especially of commensal animals [16,19,57]. It remains possible that these Polynesian chicken introductions into South America were not in high enough numbers to survive and have a genetic impact on modern-day chicken populations on the continent.

Additionally, it appears that contemporary European chickens are more allied with South Asian chickens than any other continental or insular East Asian chicken populations (table 2, AMOVA). This may suggest a more southern route to Europe from South Asia through Persia and Greece [7] rather than the northern alternative through China and Russia [4], although without any Russian samples, this is speculation only. Furthermore, the posited natural range (South Asia, Southeast Asia) of chickens retains a high level of genetic diversity (table 3).

# 4. Conclusion

The present-day global landscape of chicken genetics still appears to reflect, to some degree, the processes that allowed them to spread to regions outside the biogeographic range of their ancestors. While we can speculate that some of the original colonization patterns have been overwritten by the global commercial transport of chicken during modern times, the extent of this overwriting is uncertain. In the current study, we illustrate that modern genetic diversity of South American chickens reflects that from their

**Table 3.** Inter-population genetic diversity statistics calculated from chicken mitochondrial control region sequences from South America, Europe, Central Asia, East Asia, South Asia, Mainland and Island Southeast Asia and Pacific Islands. Number of samples (*n*), number of segregating sites (*S*), number of haplotypes (*H*), haplotype diversity (*H*$_d$) and nucleotide diversity ($\pi$).

| population | *n* | *S* | *H* | *H*$_d$ | $\pi$ |
|---|---|---|---|---|---|
| *South America* | | | | | |
| Bolivia | 11 | 12 | 6 | 0.836 | 0.01558 |
| Brazil | 47 | 20 | 9 | 0.826 | 0.03364 |
| Chile | 137 | 20 | 17 | 0.696 | 0.01390 |
| Colombia | 141 | 29 | 21 | 0.569 | 0.01335 |
| Ecuador | 82 | 30 | 20 | 0.787 | 0.01552 |
| Peru | 31 | 19 | 10 | 0.817 | 0.01914 |
| *Europe and Central Asia* | | | | | |
| Azerbaijan | 5 | 1 | 2 | 0.400 | 0.00220 |
| Europe | 32 | 24 | 14 | 0.819 | 0.02043 |
| Iberia | 73 | 16 | 6 | 0.706 | 0.01107 |
| Turkmenistan | 5 | 10 | 3 | 0.700 | 0.02198 |
| *East Asia* | | | | | |
| China | 663 | 65 | 108 | 0.943 | 0.04446 |
| Korea | 31 | 18 | 7 | 0.802 | 0.03561 |
| *South Asia* | | | | | |
| Bangladesh | 96 | 39 | 25 | 0.928 | 0.04140 |
| India | 352 | 123 | 95 | 0.886 | 0.03712 |
| *Mainland Southeast Asia* | | | | | |
| Laos | 73 | 28 | 15 | 0.805 | 0.03141 |
| Myanmar | 40 | 28 | 14 | 0.924 | 0.04895 |
| Thailand | 12 | 29 | 8 | 0.939 | 0.05960 |
| Vietnam | 161 | 47 | 41 | 0.877 | 0.03910 |
| *Island Southeast Asia* | | | | | |
| Indonesia | 625 | 50 | 90 | 0.809 | 0.01587 |
| Philippines | 259 | 49 | 57 | 0.934 | 0.02653 |
| *Pacific Islands* | | | | | |
| Fiji | 2 | 0 | 1 | 0.000 | 0.0000 |
| Solomon | 3 | 0 | 0 | 0.000 | 0.0000 |
| Vanuatu | 9 | 1 | 2 | 0.222 | 0.0012 |
| Easter Island | 4 | 0 | 1 | 0.000 | 0.0000 |

well-known Columbus-era European and Asian trading partners, rather than speculative earlier contacts with Polynesians. The genetic make-up of South American chickens is different from the genetic lineages characterizing those from the Pacific and Island Southeast Asia. Thus, based on this modern chicken diversity, no evidence of a Polynesian pre-Columbian contribution to South American chickens is observed. This does not altogether dismiss the potential for interactions between the New World and Polynesia. However, if these interactions occurred, any unambiguous evidence for it has yet to be discovered.

Finally, while modern chicken datasets can suggest hypotheses about past interactions, caution is warranted. Only securely dated and genotyped chicken material from South America, preferably pre-dating European arrival by a few centuries, can definitively rule in or out pre-Columbian Polynesian contact with South America.

Ethics. The Ethics Committee in Research of the Faculty of Agricultural Sciences of the University National of Palmira Headquarters considers that the activities proposed to carry out this research do not involve procedures against animal welfare. Therefore, the study is classified as without risk.

Data accessibility. All mtDNA control region sequences generated by this study are available in GenBank under accession numbers (MN149634-MN149862).

Authors' contributions. M.B.H. contributed to data analysis, interpretation and writing the manuscript. S.K. carried the molecular laboratory work. J.G. and V.T. contributed to data analysis, interpretation and helped in drafting the manuscript. J.G. and J.J.A. design the study. J.G. organized access to samples. J.A.A., D.Q., H.R., L.A.A., M.F.R. and H.J. assisted in the design of the study. All authors gave their approval for publication.

Competing interests. We declare we have no competing interest.

Funding. This research was funded by the Australian Research Council (ARC) Discovery Project DP110105187 to J.J.A. and J.G.

Acknowledgements. We thank the Office of the Vice Chancellor for Research and Development (OVCRD) of the University of the Philippines Diliman for the Research Load Credit given to Michael Herrera. We also thank Archaeological Studies Program's Emil Charles Robles for assisting in the preparation of the figures. Finally, Joan Tara Reyes-Hernandez and Emmanuel Jayson Bolata for some insights in the discussion.

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
