## [Reviewer comments · Royal Society Open Science]

Review History

RSOS-191558.R0 (Original submission)

Review form: Reviewer 1

Is the manuscript scientifically sound in its present form?

Yes

Are the interpretations and conclusions justified by the results?

Yes

Is the language acceptable?

Yes

Do you have any ethical concerns with this paper?

No

Have you any concerns about statistical analyses in this paper?

No

Recommendation?

Accept with minor revision (please list in comments)

Comments to the Author(s)

Herrera et al. have conducted extensive sampling of modern chickens from South America. Using mitochondrial control region sequences they showed that the majority of South American chickens were haplogroup E, with smaller contributions of haplogroup A, B and C. Interestingly there were no individuals in haplogroup D, suggesting the hypothesised pre-Columbian introduction of chickens to South America from Polynesia, if it occurred, may not have been successful. This manuscript is well written and the analyses appropriate. While not mitogenomes, the CR contains a lot of genetic variation and will provide a useful modern baseline reference dataset for further study, and is the most comprehensive study on South American chickens to date. My comments are minor:

Line 94-96: Radiocarbon dates have also been used to suggest a pre-Columbian introduction of chickens to South America, though there is also debate over the reliability of these radiocarbon dates.

Line 156: Was the TN model determined using ModelTest?

Line 186: "Five of these haplogroups (A-E) are relevant to the present study as they are found in South American chickens". Haplogroup D has not been found in modern South American chickens in this study.

Figure 1: I know there is a lot of genetic variation in chicken mitochondrial CR, but is there a nicer/cleaner way to show this haplotype network?

Review form: Reviewer 2

Is the manuscript scientifically sound in its present form?

Yes

Are the interpretations and conclusions justified by the results?

Yes

Is the language acceptable?

Yes

Do you have any ethical concerns with this paper?

No

Have you any concerns about statistical analyses in this paper?

No

Recommendation?

Accept with minor revision (please list in comments)

Comments to the Author(s)

This manuscript is part of a large debate of whether Polynesian chickens were introduced to the Americas in Pre-Columbian times. I believe this study will be of great importance to help settle this debate. The manuscript is well written and concise. I do not have expertise in haplotype

analyses, so I cannot comment on the analytical methods used. However, I did notice the GenBank Accession numbers provided don't seem to yield results. Please double check this.

Decision letter (RSOS-191558.R0)

04-Nov-2019

Dear Dr Herrera

On behalf of the Editors, I am pleased to inform you that your Manuscript RSOS-191558 entitled "European and Asian contribution to the genetic diversity of mainland South American chickens" has been accepted for publication in Royal Society Open Science subject to minor revision in accordance with the referee suggestions. Please find the referees' comments at the end of this email.

Both the reviewers consider this an important manuscript and have recommended publication, but also suggest some minor revisions to your manuscript. Therefore, I invite you to respond to the comments and revise your manuscript.

- Ethics statement

- Data accessibility

If you wish to submit your supporting data or code to Dryad (<http://datadryad.org/>), or modify your current submission to dryad, please use the following link:
<http://datadryad.org/submit?journalID=RSOS&manu=RSOS-191558>

- Competing interests

- Authors' contributions

- Acknowledgements

- Funding statement

Because the schedule for publication is very tight, it is a condition of publication that you submit the revised version of your manuscript before 13-Nov-2019. Please note that the revision deadline will expire at 00.00am on this date. If you do not think you will be able to meet this date please let me know immediately.

- 1) A text file of the manuscript (tex, txt, rtf, docx or doc), references, tables (including captions) and figure captions. Do not upload a PDF as your "Main Document";
- 2) A separate electronic file of each figure (EPS or print-quality PDF preferred (either format should be produced directly from original creation package), or original software format);
- 3) Included a 100 word media summary of your paper when requested at submission. Please ensure you have entered correct contact details (email, institution and telephone) in your user account;

- 4) Included the raw data to support the claims made in your paper. You can either include your data as electronic supplementary material or upload to a repository and include the relevant doi within your manuscript. Make sure it is clear in your data accessibility statement how the data can be accessed;
- 5) All supplementary materials accompanying an accepted article will be treated as in their final form. Note that the Royal Society will neither edit nor typeset supplementary material and it will be hosted as provided. Please ensure that the supplementary material includes the paper details where possible (authors, article title, journal name).

Kind regards,
Andrew Dunn
Senior Publishing Editor
Royal Society Open Science
openscience@royalsociety.org

on behalf of Dr Steve Brown Steve Brown (Subject Editor)
openscience@royalsociety.org

Reviewer comments to Author:
Reviewer: 1

Comments to the Author(s)

Herrera et al. have conducted extensive sampling of modern chickens from South America. Using mitochondrial control region sequences they showed that the majority of South American chickens were haplogroup E, with smaller contributions of haplogroup A, B and C. Interestingly there were no individuals in haplogroup D, suggesting the hypothesised pre-Columbian introduction of chickens to South America from Polynesia, if it occurred, may not have been successful. This manuscript is well written and the analyses appropriate. While not mitogenomes, the CR contains a lot of genetic variation and will provide a useful modern baseline reference dataset for further study, and is the most comprehensive study on South American chickens to date. My comments are minor:

Line 94-96: Radiocarbon dates have also been used to suggest a pre-Columbian introduction of chickens to South America, though there is also debate over the reliability of these radiocarbon dates.

Line 156: Was the TN model determined using ModelTest?

Line 186: "Five of these haplogroups (A-E) are relevant to the present study as they are found in South American chickens". Haplogroup D has not been found in modern South American chickens in this study.

Figure 1: I know there is a lot of genetic variation in chicken mitochondrial CR, but is there a nicer/cleaner way to show this haplotype network?

Reviewer: 2

Comments to the Author(s)

This manuscript is part of a large debate of whether Polynesian chickens were introduced to the Americas in Pre-Columbian times. I believe this study will be of great importance to help settle this debate. The manuscript is well written and concise. I do not have expertise in haplotype analyses, so I cannot comment on the analytical methods used. However, I did notice the GenBank Accession numbers provided don't seem to yield results. Please double check this.

Author's Response to Decision Letter for (RSOS-191558.R0)

See Appendix A.

Decision letter (RSOS-191558.R1)

02-Jan-2020

Dear Dr Herrera,

It is a pleasure to accept your manuscript entitled "European and Asian contribution to the genetic diversity of mainland South American chickens" in its current form for publication in Royal Society Open Science.

on behalf of Prof Steve Brown (Subject Editor)
openscience@royalsociety.org

Appendix A

Manuscript RSOS-191558: European and Asian contribution to the genetic diversity of mainland South American chickens

RESPONSES TO REFEREES:

Reviewer comments to Author:

Reviewer: 1

Comments to the Author(s)

Herrera et al. have conducted extensive sampling of modern chickens from South America. Using mitochondrial control region sequences they showed that the majority of South American chickens were haplogroup E, with smaller contributions of haplogroup A, B and C. Interestingly there were no individuals in haplogroup D, suggesting the hypothesised pre-Columbian introduction of chickens to South America from Polynesia, if it occurred, may not have been successful. This manuscript is well written and the analyses appropriate. While not mitogenomes, the CR contains a lot of genetic variation and will provide a useful modern baseline reference dataset for further study, and is the most comprehensive study on South American chickens to date. My comments are minor:

Line 94-96: Radiocarbon dates have also been used to suggest a pre-Columbian introduction of chickens to South America, though there is also debate over the reliability of these radiocarbon dates.

RESPONSE: The suggestion has been integrated. But statement in lines 100-102 was removed as it conveys the same information.

Line 156: Was the TN model determined using ModelTest?

RESPONSE: jModelTest was used to determine the model that best fit the data.

“jModelTest⁴⁷ was used to determine the best-fit model for the CR dataset (TIM1+G), then a NJ tree was estimated using PAUP v.4⁴⁸”.

Line 186: "Five of these haplogroups (A-E) are relevant to the present study as they are found in South American chickens". Haplogroup D has not been found in modern South American chickens in this study.

RESPONSE: Referee suggestion has been integrated into the manuscript in Lines 187-188.

Figure 1: I know there is a lot of genetic variation in chicken mitochondrial CR, but is there a nicer/cleaner way to show this haplotype network?

RESPONSE: I am afraid that this the cleanest haplotype network that we can manage given this dataset. We have tried various network ordering software before, but this is clearest.

Reviewer: 2

Comments to the Author(s)

This manuscript is part of a large debate of whether Polynesian chickens were introduced to the Americas in Pre-Columbian times. I believe this study will be of great importance to help settle this debate. The manuscript is well written and concise. I do not have expertise in haplotype analyses, so I cannot comment on the analytical methods used. However, I did notice the GenBank Accession numbers provided don't seem to yield results. Please double check this.

RESPONSE: DNA sequence data were submitted to GenBank via BankIt224193 and given the accession numbers MN149634-MN149862. However, they will not be released to the public database until 5 July 2020 or until the data or accession numbers appear in print, whichever is first.

Other changes:

1. I have modified and added another affiliation for author Han Jianlin.
2. Added an ethics statement.